# Indian Biosimilars and Vaccines at Crossroads–Replicating the Success of Pharmagenerics

**DOI:** 10.3390/vaccines11010110

**Published:** 2023-01-02

**Authors:** Sunita Panda, Puneet Kumar Singh, Snehasish Mishra, Sagnik Mitra, Priyabrata Pattnaik, Sanjib Das Adhikary, Ranjan K. Mohapatra

**Affiliations:** 1School of Biotechnology, KIIT Deemed University, Bhubaneswar 751024, India; 2Department of Biosciences and Biomedical Engineering, Indian Institute of Technology, Indore 453552, India; 3Merck Pte Ltd., 2 Science Park Drive, Ascent Building, #05-01/12, Singapore 118222, Singapore; 4Department of Anesthesiology and Perioperative Medicine, Penn State College of Medicine, Hershey, PA 17033, USA; 5Department of Chemistry, Government College of Engineering, Keonjhar 758002, India

**Keywords:** Indian biosimilars, pharmagenerics, healthcare scenario, human resource, visionary perspective

## Abstract

Background: The global pharma sector is fast shifting from generics to biologics and biosimilars with the first approval in Europe in 2006 followed by US approval in 2015. In the form of Hepatitis B vaccine, India saw its first recombinant biologics approval in 2000. Around 20% of generic medications and 62% of vaccines are now supplied by the Indian pharmaceutical industry. It is this good position in biologics and biosimilars production that could potentially improve healthcare via decreased treatment cost. India has witnessed large investments in biosimilars over the years. Numerous India-bred new players, e.g., Enzene Biosciences Ltd., are keen on biosimilars and have joined the race alongside the emerging giants, e.g., Biocon and Dr. Reddy’s. A very positive sign was the remarkable disposition during the COVID-19 pandemic by Bharat Biotech and the Serum Institute of India. India’s biopharmaceutical industry has been instrumental in producing and supplying preventives and therapeutics to fight COVID-19. Despite a weak supply chain and workforce pressure, the production was augmented to provide reasonably priced high-quality medications to more than 133 nations. Biosimilars could cost-effectively treat chronic diseases involving expensive conventional therapies, including diabetes, respiratory ailments, cancer, and connective tissue diseases. Biologics and biosimilars have been and are being tested to treat and manage COVID-19 symptoms characterized by inflammation and respiratory distress. Purpose of review: Although India boasts many universities, research centers, and a relatively skilled workforce, its global University–Industry collaboration ranking is 24, IPR ranking remains 47 and innovation ranking 39. This reveals a wide industry-academia gap to bridge. There are gaps in effective translational research in India that must be promptly and appropriately addressed. Innovation demands strong and effective collaborations among universities, techno-incubators, and industries. Methodology: Many successful research findings in academia do not get translation opportunities supposedly due to low industrial collaboration, low IP knowledge, and publication pressure with stringent timelines. In light of this, a detailed review of literature, including policy papers, government initiatives, and corporate reviews, was carried out, and the compilation and synthesis of the secondary data were meticulously summarized for the easy comprehension of the facts and roadmap ahead. For easy comprehension, charts, figures, and compiled tables are presented. Results: This review assesses India’s situation in the biosimilar space, the gaps and areas to improve for Indian investment strategies, development, and innovation, addressing need for a more skilled workforce, industrial collaboration, and business models. Conclusions: This review also proposes forward an approach to empowering technopreneurs to develop MSMEs for large-scale operations to support India in taking innovative thoughts to the global level to ultimately realize a self-reliant India. The limitations of the compilation are also highlighted towards the end.

## 1. Introduction

Due to the high cost of biologics, new legislation is introduced to encourage the development of biosimilars, which will increase treatment options, broaden access, and reduce costs. Biosimilars are structurally similar to their original biological molecule (originator; inspirer; reference product) but are not identical [1,2,3]. Nevertheless, their bioactivity is clinically very similar to the reference product in terms of safety and efficacy [4,5]. Their medication and function are similar to the original biologic at the same strength and dosage. Manufacturers of biosimilars could obtain commercial authorization to market them as biological drugs, being remarkably similar, after resolving the regulatory exclusivity and intellectual property issues for a biologic. There is an increasing demand for biologics and biosimilars particularly in treating auto-immune diseases and cancer (https://www.pfizer.com/news/articles/biologics_vs_biosimilars_key_differences_explained; 7 September 2022). The prevailing pandemic has severely dented healthcare and demanded vaccines, biologics, biosimilars, and generics to prevent or manage the symptoms.

There are a lot of competitions to develop, manufacture, market, and approve biologics and biosimilars globally. The COVID-19 pandemic demanded even monoclonal antibodies (mAbs), e.g., tocilizumab, sarilumab, and itolizumab, as therapeutics [6,7,8,9]. As a silver lining in the dark cloud, the pandemic has marked the turning point for Indian pharmaceuticals to put its best foot forward. The Indian government is encouraging domestic bioindustries to transform the crisis that COVID-19 poses into an opportunity to spread their wings in the biosimilar market. Indian vaccine stories shined well during the crisis period. New business models and approaches to develop indigenous vaccines, generics, and biosimilars were commendable. For instance, Zydus Cadila (a concern of Cadila healthcare) focused on interferon *α*-2b for commercialization as a biosimilar version for treatment (https://adisinsight.springer.com/drugs/800040283; 7 September 2022). Indian biopharma companies have shaped up the vaccine and biosimilar landscape with the entry of two additional Indian entities into the global marketplace during the pandemic. India also competes for global market positioning by adding strategies targeting the booming clinical trials and clinical research, increasing its skilled workforce, adopting innovative techniques in manufacturing, etc. Due to all this path-breaking, multipronged evolution, there is a need for revamping Indian regulatory standards for biosimilars approval.

The global demand for vaccines and biosimilars is on the rise, and therefore there is a huge market opportunity for biopharma business. A meticulous step-by-step plan for collaborations, assured prompt regulatory approval, and optimal market access for biosimilar research and commercialization are necessary—an indescribably complex endeavor. In face of the heightened competition from biosimilars, the businesses attempting to preserve the market share of branded biologic products are losing market focus. In order to maintain the momentum in the given circumstances, an integrated, strategic development and commercial plan from the very beginning of any biosimilar development programme could ensure the distinct competitiveness of biosimilars. This compilation intends to provide insights on the strengths, drawbacks, opportunities, and challenges of Indian biotech and biopharma companies as global players. The article discusses the history, scope, and evolution of biotechnology companies in India, methodologies, global vs. Indian biotech endeavors, market size and current scenario of Indian biotech companies in biopharma sector, healthcare scenario, translational research initiatives, prospects and bottlenecks, what ails Indian biotech going global, Indian vaccines in the global market, visionary perspectives, and ending with a conclusion. It emphasizes elevated Indian strategies and participation to lead and regulate the international biosimilars and vaccines market.

## 2. History, Scope and Evolution of Biotechnology Companies in India

Biotechnology is the development of products using biological systems (living organisms or their derivatives), including recombinants engineered from wild types to improve human life [10,11]. Its roots are in ancient processes that used animals, plants, and microorganisms to produce a variety of goods with economic benefits for humans. Modern biotechnology uses molecular-level genetic engineering to create transgenic (genetically modified) organisms that can be used to create vaccines, drugs, and diagnostic tools, among other things. Contrary to GM crops, which increase production to feed the world’s population and produce biofuels for a cleaner environment, biopharmaceuticals, such as biologics and biosimilars, offer treatments for many fatal diseases which are critical for disease-free survival.

Therefore, the Indian government is aggressively promoting the development of the biotech sector due to its socioeconomic influence and potential for a sustainable future. India’s bio-based industries are booming because of financial support from both the public and commercial sectors (such as angel investors and venture capitalists). Start-up grants, investments, and many other internal and external forms of financial support have been drawn to high-end innovative start-ups, particularly by recent Indian university graduates [12]. Indian bioindustry comprises of five broad divisions: biopharmaceuticals [13], agribiotech [14], bioinformatics [15], fermentation, and bio-based services. Producing medicines, diagnostics, and vaccinations (which are preventives) are all part of the biopharmaceutical industry. Of all bioindustries, the Indian biopharma sector is the one that generates the most income. The third-largest industry is agribiotech, which deals with transgenic plants, hybrid seeds, biofertilisers, biopesticides, etc. In the most recent period (2010–2011), agribiotech’s revenue share climbed by 14%. Utilizing cells or cell derivatives, e.g., protein, natural amino acids, and yeast, bio fermentation (also known as white biotechnology) produces chemicals that are generally used in a B-to-B mode in the starch, refinery, liquor, materials, textile, and leather (tanning) sectors, to mention a few. Bio services, on the other hand, represent the only discernible bio industrial sector in India that deals with services, e.g., clinical trials, contract research, trading, etc., rather than necessarily involving a tangible product (https://www.indianmirror.com/indian-industries/biotechnology.html; https://www.birac.nic.in/; 7 September 2022) (Figure 1a,b).

Indian biotechnology and its industrial revolution present a significant historical legacy [16,17]. The Centre for Cellular and Molecular Biology (CCMB) set up by the Council of Scientific and Industrial Research (CSIR) was the first institution devoted to biotechnology in India in 1977. Subsequently, the National Institute of Immunology (NII) was founded by the Department of Science and Technology (DST) to support advanced biological research. The National Biotechnology Board (NBTB) was established in 1982 to promote scientific programmes in biotechnology and to strengthen indigenous capabilities. The NBTB was upgraded to an autonomous body in 1986 as the Department of Biotechnology (DBT) under the Ministry of Science and Technology who planned to promote and coordinate biotechnology programmes. DBT focused on improving scientific research both quantitatively and qualitatively, providing appropriate infrastructure, utilizing human resources, and promoting industry–academia collaborations [18,19]. The DBT developed the National Institute of Plant Genome Research (NIPGR), The Centre of DNA Fingerprinting and Diagnostics (CDFD), National Brain Research Institute (NBRI), Institute of Bioresources and Sustainable Development (IBSD), Institute of Life Sciences (ILS), Institute for Stem Cell Biology and Regenerative Medicine (INSTEM), National Agri-Food Biotechnology Institute (NABI), Translational Health Science and Technology Institute (THSTI), National Institute of Biomedical Genomics (NIBMG), etc. The DBT has also established biotechnology parks and bioincubators in various states, e.g., Uttar Pradesh (Lucknow), Karnataka (Bengaluru), Telangana (Hyderabad), Tamil Nadu (Chennai), Odisha (Bhubaneswar), Kerala, and Assam (https://dbtindia.gov.in/; https://dbtindia.gov.in/about-us/organization-structure/public-sector-undertaking, 7 September 2022). They are successfully accelerating the commercialization of new biobased technologies, offering facilities to scientists, small and medium enterprises and promoting public-private partnerships. DBT established another autonomous unit, Biotechnology Industry Research Assistance Council (BIRAC), to promote innovation, empower emerging biotech start-ups, commercialize innovative discoveries, and promote industry–academia interactions [20].

The foundation and growth of the Indian biotech industry is stimulated by different factors in the post-independence era [21]. Indian biotech ventures were established by various entrepreneurs working in Indian industrial set-ups and pharma industries, or academic scientists with industrial experience, or the home-coming of Indian scientists, post docs, and entrepreneurs, returning with vast industrial or academic experience. New biotech set-ups were founded by extending a division from the existing drug industry or diversifying a non-pharma company into a biotech start-up or established by multinational companies (Table 1). Government initiatives and innovative start-ups are involved in the generation of bio industries, involving diagnostics, bioinformatics, regenerative medicines, generics, vaccines, and biotherapeutics.

Bioindustry start-ups give India a place in the top 12 global destinations and 3rd place in Asia after China and Japan [22]. As per the Association of Biotechnology Led Enterprises (ABLE), the biotech sector will cross US $100 billion by 2025. With the biopharma sector having the largest share among the biotech set-up in India, it stands as the 3rd largest producer of pharmaceuticals globally and holds 14th rank by value (https://www.ibef.org/industry/pharmaceutical-india, 7 September 2022). While India has a stronghold in global market by supplying 40% of generics, more than 50% of global demand of vaccines is fulfilled by Indian pharma sector. The export of pharmaceuticals from India in the FY20 was US $16.3 billion. India’s domestic pharmaceutical market turnover reached US $20.03 billion in 2019, a 9.8% increment from US $18.12 billion in 2018. The pharmaceutical export market turnover was US $24.4 billion in 2020–2021, witnessing an 18.1% (YoY) growth. India’s domestic pharmaceutical market is estimated at US $41 billion in 2021, likely to grow to US $65 billion by 2024, and further expected to reach US $130 billion by 2030 (https://www.thehindu.com/brandhub/the-giant-leap-of-indias-pharmaceutical-industry/article65670866.ece, 7 September 2022).

The global pharma sector is fast shifting from generics to biologics and biosimilars. Biosimilars hold potential to improve patient’s life by decreasing the treatment cost (https://www.gabionline.net/biosimilars/general, 7 September 2022). India is well placed in biologics and biosimilars production. Over the years, India has seen large investments in biosimilars [23]. Apart from Indian-bred indigenous companies, e.g., Biocon and Dr. Reddy’s, who are currently the emerging giants, numerous new entities, e.g., Enzene Biosciences Ltd. (a subsidiary of Alkem Laboratories), are keen to produce biosimilars and have joined the race. Some of the other Indian biopharma giants have been briefly discussed in Appendix A. The first approval of biosimilars was in 2006 in Europe, followed by the US in 2015. Since then, several biosimilars have been developed that are cost-effective and utilised across the globe to treat chronic diseases for their affordability and quality. They have also been used to treat non-infectious diseases, e.g., diabetes, respiratory problems, cancer, and connective tissue diseases, etc. India got its first approval for a Hepatitis B vaccine in 2000 [24]. The biosimilar guidelines were established in 2012 jointly by the Central Drug Standard Control Organisation (CDSCO) and the Department of Biotechnology (DBT). These guidelines on the production and approval of biosimilars were further revised in 2016. In 2019, India had 98 approved biosimilars. Currently, a couple of hundred active biosimilars are at various stages of research and development by several Indian biopharma industries [25].

## 3. Methodology

Data were sourced from the reports and publications available in the public domain, compiled, analyzed, and synthesized. Data were collected from authentic databases, e.g., Scopus, ScienceDirect, PubMed, Web-of-Science, and GoogleScholar, to name a few, information in the public domain (the websites) on several public and government health organizations and line departments, as well as policy making bodies. Numerous publicly available policy documents on the pharma companies’ profile and several reference books were also consulted. For critical in-depth coverage and compilation of the relevant contents, the key terms searched online included Indian biosimilars, Indian biopharmaceutical companies, growth of Indian bioindustries, biotech companies in the global biosimilars market, vaccine manufacturers, pharmagenerics, healthcare scenario, human resource inventories, etc. All the data thus obtained were carefully examined and only the closely matched reports/studies were considered for compilation and critical discussion while excluding the irrelevant or generalized reports. More than 300 reports and research documents were interpreted, 170 of which fulfilled the objective of the study and were discussed. All such documents have been duly cited and listed in the references section.

## 4. Global vs. Indian Biotech Endeavors

Research conducted by the Market line and DCAT on the global ranking of countries in terms of their biotechnological innovations states that the US is dominating with 48.2% in biotech market, 24% of the share is held by Asia-Pacific, followed by the Europe with 18.1%. The Middle East contributes 1.8% while the remaining 7.9% of the market is captured by the rest of the world. India carries 3% of share in the global market and is placed at 52nd position in global ranking (https://birac.nic.in/big.php, 7 September 2022). Further, India is the pioneer in the worldwide supply of DPT, BCG, and measles vaccines. Biotechnology has the potential to drive India’s economy to USD 5 Trillion by 2024. The US and the European Union are leading in this sector [26]. The US is at the leading position followed by Japan with respect to R&D expenditure in the biopharmaceutical space. Other countries, e.g., Switzerland, Germany, France, and Denmark, also significantly increased their investment in R&D in the recent past. China boosted its R&D expenses by 9.1% from 2014 to 2018 (https://www.gabionline.net/biosimilars/general, 7 September 2022). As indicated by a report of the Organization for Economic Cooperation and Development (OECD) on science and innovation (2010), the business picture has also improved for a few non-OECD nations, including Brazil, India, China, Singapore, and South Africa [27].

Europe is leading in terms of global production and commercialization of biosimilars, followed by Asia-Pacific nations which include countries, such as South Korea, Japan, India, and China [28]. While Korea contributes 43% to the global biosimilars market, the Indian biopharma sector is large and is in the most advanced stage to lead the biosimilars market in the Asia-Pacific. The global biosimilar market is expected to reach US $35.7 billion by 2025 from the current US $11.8 billion (2020) and expected to grow at a CAGR of 24.7% [27].

A list of manufactured biosimilars have also been provided to understand how other countries have fared in the biosimilars market capture (Table 2) along with Indian biosimilars that have been provided in Table 3. India launched its first rituximab biosimilar Reditux by Dr. Reddy’s Laboratories. The world’s first adalimumab biosimilar Exemptia was manufactured and marketed by Zydus Cadila in India [29]. CANMab, the world’s first biosimilar version of Herceptin, developed collaboratively by Biocon and Mylan [30], was introduced in India in Feb 2014. Biocon launched a biosimilar Glargine Insulin in 2016, and successfully marketed the same in Japan. Biocon and Mylan as global partners developed Ogivri, Trastuzumab, and Fulphila (Pegfilgrastim), which have received US FDA approval [24]. All these achievements speak highly to how India has made its name in the biosimilars market with respect to the other countries.

## 5. Market size and Current Scenario of Indian Biotech Companies in Biopharma Sector

Bioindustry has offered boost to the Indian economy. There are more than 2500 Indian biotechnology companies and more than 2700 start-ups. Indian bioindustry was worth US $63 billion in 2019, and is projected to be US $102 billion by 2025, with a CAGR of 10.9%. During this period, it is expected to spike by 19% from the current 3% of global market share. Biopharmaceuticals alone contributed about 62% in 2020, trailing 16% behind bioagriculture and 15% behind bioservices [58,59]. The sector is growing and becoming a leading clinical trial, contract research, and manufacturing destination. The Indian pharma sector inclines towards research and production of particularly biosimilars and biologics. Research works in India on biosimilars may provide promising cost-effective therapeutic solutions especially during recent times when the globe is dealing with recovery from the COVID-19 pandemic. The major current players in the Indian biosimilars market are briefly described in Appendix A.

India is the second worst hit country in terms of health and economics by the recent COVID-19 pandemic, affecting various sectors, e.g., food, agriculture, aviation, and tourism, because of which the GDP has caved in [60,61,62]. The Indian pharmaceutical industry struggled because of the bans imposed on the import/export of a few crucial drugs, equipment, and PPE kit. However, the ‘*Atmanirbhar Bharat*’ (Self-reliant India) initiative offered great incentive in enhancing economic activities. Poor accessibility to raw materials due to a disrupted supply chain made it harder to meet the increasing demand for drugs. Despite the economic crises, labor shortage, and logistics crisis, the Indian pharma industry has raised hope and is constantly pursuing the development of generic drugs and vaccines (Table 4). The country is currently dominating the global generics market with a size of US $55 billion. Covishield, being manufactured by the Serum Institute of India, was developed jointly by Astrazeneca and Oxford University. Covaxin is the first indigenous vaccine by Bharat Biotech Ltd. in collaboration with the Indian council of Medical Research (ICMR) and the National Institute of Virology (NIV). Both the vaccines have been approved for ‘emergency use’ by the Drug Controller General of India (DCGI). Other vaccines either received emergency authorization or remain under different phases of clinical trials, as detailed in Table 5.

Various biologics and biosimilars have been tested and are under trial to treat and manage COVID-19 symptoms characterized by inflammation and respiratory distress, e.g., Celltrion Healthcare’s infliximab biosimilar, Remsima, to control cytokine release syndrome mediated immune response in COVID-19, and Bevacizumab, to treat pulmonary oedema related acute lung injury (ALI) and acute respiratory distress syndrome (ARDS). Tocilizumab, an anti–IL-6 receptor monoclonal antibody (mAb) has been proven to treat COVID-19 by reducing inflammation and maintaining blood pressure levels. Eculizumab and Ravulizumab are under clinical trials to treat moderate to severe pneumonia-associated COVID-19. Itolizumab developed by Biocon biologics is also used to treat chronic respiratory syndrome (CRS) in COVID-19 patients, approved for emergency use by Indian regulatory authorities. Pfizer evaluating a biologic tofacitinib to treat inflammation, and Zydus Cadila is investigating on the efficacy of Interferon *α*-2b to treat COVID-19 [69,70]. Apart from the already established pharma and biopharma giants, the Indian government along with DBT and other foreign collaborators such as the World Bank are working on providing various grants and funding schemes to bridge the gap between industry and academia, as detailed in Appendix A.

## 6. Healthcare Scenario, Translational Research—Initiatives, Prospects and Bottlenecks

Translational research is an interdependent lab-to-land or bed to bench scale process by which the findings of basic research are translated into economically feasible applications to improve or solve problems in healthcare, ecology and ecosystem, and other identified sectors. Translational research ensures progress in basic scientific research and the application of the scientific understanding and technological advancements for economic benefits. Translational research has two phases, T1: basic science discoveries utilized to develop novel processes or product, and T2: clinical research focusing on improving the existing clinical practices [71,72]. Having witnessed a great rise as a key global player in generic pharmaceuticals, it is observed that baring a few instances here-and-there India faces various challenges in the research and development of novel clinical interventions, especially on account of serious long-term sustainable collaborations with the key global players. This is slowly but steadily eroding the much-awaited organic growth in this promising sector with strong economic potential.

The Indian government through DBT under the Ministry of Science and Technology, has taken several initiatives to promote translational research in India. The first initiative was to establish an autonomous institution ‘Translational Health Science and Technology Institute’ in 2010. The focus of the organization is to develop novel strategies for the diagnosis and management of diseases by promoting innovative healthcare through multidisciplinary approach through inter-institutional and industry–academia collaborations to promote translational research.

The Stanford-India Biodesign (SIB) programme was started under its aegis in 2007 and continued till 2014. SIB was an innovation programme implemented by Stanford University, DBT, All India Institute of Medical Sciences (AIIMS), New Delhi, and Indian Institute of Technology, Delhi (IITD), in collaboration with Queensland University of Technology, Australia and Hiroshima University, Japan. The objective was to train the medical technology innovators in India to innovate processes and products that are affordable and accessible to the Indian population. The programme trained more than 100 medical technology innovators and entrepreneurs, more than 50 prototypes were developed, and more than 50 medical devices were developed by the young innovators. Of these, 15 technologies have been translated successfully, including five medical devices, and 11 start-ups became sustainable [73].

National Pharma mission through the ‘Make in India’ campaign had also empowered start-ups, enterprises, and increased industry-academia collaborations. The Central Drug Standard Control Organization (CDSCO), India, is also trying to reform the rules for approval process for novel drugs, new interventions, and new medical devices as per the prevailing requirements [73].

The goal of translational research is doing work in this biotech sector. However, there are certain roadblocks, bottlenecks, and issues at various levels, primarily societal, cultural, and regulatory. The first and important challenge is the difference of approach between the clinicians and the general scientists. The differences lead to a lack of communication, training, and education, and thereby varied views on the research and development outcomes. The other critical factor could be the cultural difference between the nations. The lack of appropriate scientific infrastructure, resources and workforce also is a critical barrier for translational research. The next important factor is the complex regulatory environment and ethical issues, including intellectual property rights, cell/tissue banking, sample transfer regulations, toxicology (safety issues), product manufacturing regulations, approval process for the product before hitting the market, clinical trials, etc. [74].

Another important issue in translational research is related to funding and incentives provided to research scientists. Governmental funding mechanism is time-consuming to obtain and non-transparent. The review process for a grant is longer, and the tracking of grants is not accessible. The communication process from reviewer to investigator is not clear. These kinds of issues need attention to improve the culture of scientific research and its output. Scientists, clinical researchers, and academicians are demanding a transparent, duplication-free, improved Indian grant framework. A common online portal for grant applications will reduce the time for grant revision, maintenance of transparency, and decrease the duplication of scientific objectives. Some researchers are also pointing towards a lack of incentives and rewards to researchers. A thorough review process should be implicated, and merit-based award systems must be created to provide support and encouragement to the scientific community. These funding agencies should have both academic and industrial experts for the neat examination of research objectives with translational potential. Another important solution to improve Indian research funding is to build a national mission mode involving public bodies, different trusts, and private institutes [75].

## 7. What Ails Indian Biotech Going Global?

The notion that developing nations are, at best, good service providers and lack the infrastructure to participate in cutting-edge technology to innovate products appears to have greatly hindered India’s organic growth as compared to that of many other otherwise successful nations. This entails implementing novel approaches or methods to deal with diverse societal concerns. After the United States and China, India has the third-highest number of start-ups. Entrepreneurship-based organizations place an undue emphasis on customer acquisition at the expense of long-term revenue generation plans. In the upcoming sections below, we’ve attempted to highlight some of the potential reasons why the Indian biotech industry is struggling to expand internationally. The same is represented below as a SWOT analysis.

### 7.1. SWOT Analysis of Indian Biotechnology Industry in General and Biosimilars in Particular

As discussed, biosimilars represent the generic version of biologic products that could reduce healthcare costs. These innovated biological compounds exhibit similar efficacy and safety levels as their reference products. Thus, it provides wide-spectrum and cost-effective access to life-saving medications. However, considering the scale of operations at stake, a critical understanding of the state-of-the-art of the Indian biotech sector in general and the biosimilar industry in particular, considering the strengths, weaknesses, opportunities, and threats, is necessary. It is noteworthy that the strengths and weaknesses are the internalized factors of the biosimilar industry ecosystem and opportunities and threats are the externalized factors. Similarly, strengths and opportunities together would facilitate an organic growth to the sector while weaknesses and threats would drag the growth prospects (Figure 2).

Table 6 outlines the factors that could either positively or negatively impact the growth prospects of the Indian biosimilars and vaccines industry, which is at a crossroads in its quest to replicate the success of pharmagenerics.

### 7.2. Lack of Innovation

Although the culture of entrepreneurship has been hitting hard, start-ups invariably lack innovation and have weak business models, focusing primarily on the predictable imitative business models and not venturing into high financial risk. The number of patents filed by India is very low, and only 7% of them are filed by the start-ups. The reasons for the lack of innovation in business model are poor development, less translation-oriented education, fear of failure, and funding issues [76]. Innovation depends upon resourceful human capital, investments, business environment, and performance enablers. The government of India should establish dedicated institutions, universities, and centers of excellence to provide education on the issues pertaining innovations in SMEs to improve and promote the culture of innovation. Investments for SMEs should be continual from government and market sectors to meet the demands. Low interest loans should be provided by the banks for setting up innovative ideas. Setting up collaborations between SME clusters and government to create value across the value chain, innovators must discuss with experts and consultants to understand the consumer demands and market trends. Mindset and cultural aspect also play a major role in entrepreneurship in India. Innovators should understand that frugal engineering is not the task to improve but rather to implicate in solving the risk–reward equation for entrepreneurship in India. Indian biopharma companies are now at the early stage of innovative policies. Such innovators are working in local and international collaboration among research institutes, health organizations, universities, and Indian or foreign based firms to adopt new technologies and business policies [77].

### 7.3. Lack of Ventures

India lacks sufficient funding compared to the west for translational research. It depends largely on venture/angel funding. Due to high risk and challenging journey of innovative research, investors have reservations in supporting a start-up. The number of investors from private sectors coming forward as funders is very low. Indian government, thus, is the primary investor for start-ups [78]. However, it spends only 0.7% of its GDP in research and development while China spends 2.19%, Japan around 3.9%, South Korea spends 4.81%, and Israel spends 4.95% of its GDP in the R&D sector [79]. India is depending on overseas investors, basically from North America, Europe, Japan, and China. There is a lack of participation by domestic investors due to insufficient funds and a lack of risk-taking attitude. Another reason could be due to the regulatory environment. The difference in venture tax rates between publicly listed and private listed start-ups is discouraging the angel funding from India. Private investors and especially public banks are not interested in investing or lending commercial loans due to debt overhang in a large number of private firms [80]. Although case-specific funding from the government is appreciable, it covers only seed capital and a meagre risk capital. After proof-of-concept stage, the start-ups need accelerator funding for the commercialization stage. The commercialization of biotechnology products is longer, costlier, and risky primarily due to the legal and ethical challenges, which Indian society is typically averse to. Further, venture capitalists do not find investing in biotechnology attractive as there is no smart exit route for them. They may be more willing to make the investment if they can get a return through the IPO (stock market) [81]. However, there is a silver lining: amidst economic crises due to the recent pandemic, the year 2020 saw a five-fold increase in venture funding to the pharmaceutical sector compared to just the previous year.

### 7.4. Loose Ignition Grant System

Government funding for young professionals in India to develop through ignition grants has been respectable in recent years. However, it is urgently necessary for other sources to take comparable actions. Although there are several programmes for government support, the procedure is drawn out, requires a ton of paperwork, and requires administrative work that is out of step with the idea of ‘ease of doing business’. Additionally, it appears that the examination of a grant request is ineffective due to insufficient communication with the principal investigator. A grant proposal’s duration from submission to final approval is agonizingly protracted [82,83,84]. The cash bottleneck with the Indian funding agencies appears to be serious, even after technical approval, financial evaluation, and funding. Despite a rigorous assessment, the grant is accepted, but scientists do not get the money in a timely manner, which affects their long-term career prospects, their ambition, and motivation. Young biotechnology professionals lose momentum when they do not receive opportunities and incentives in a timely manner. It is essential to concentrate on large-scale collaboration networks, including several institutions and research groups with interdisciplinary perspectives, that could draw both local and foreign financing systems and enhance innovation [85].

### 7.5. Lack of Leadership Vision

A clear vision and good leadership are essential components that support the creation and maintenance of collaborations for the systemic, natural expansion of any organization. It is important to note that strong partnerships and long-term collaborations between businesses, institutions of higher learning, governmental organizations, and researchers are on the horizon, particularly in the biopharma industry. The creation of a start-up ecosystem is hampered by visionless leadership and assistance focused on skilled labor. As a result, 50% of start-up enterprises fail to secure funding due to a lack of capable and devoted leadership [86,87]. New businesses fail or have delayed growth because of a lack of leadership. Failures are mostly caused by the absence of long-term company objectives, the absence of creative ideas, and the slow acceptance of new business models and markets. Poor management and communication abilities also affect networking and fund-raising efforts. A start-up’s ability to scale up and grow generally is impacted by collaboration issues. According to the MSME, there are 6.33 crore small, medium, and micro-enterprises in India, with 90% of them being micro-enterprises. The workforce is mostly engaged in product manufacture, marketing, and financial management in these small and micro-enterprises. However, innovative plans to upgrade and expand the business are low. Indian entrepreneurs need to be trained to effectively manage marketing, communication, and working capital to get a holistic view on how to run a business in this kind of domain [88].

### 7.6. Quality Human Resource

The lack of a competent skilled human resource is another major concern for a start-up. *Aspiring Minds* magazine reports that only 3.84% of graduates have the basic technical and analytical skill sets needed for a start-up.

To enhance their capabilities and improve their global competitiveness, Indian companies would need highly trained personnel. Many highly skilled research scientists and PhDs migrate abroad for better financial support for their research and self. India needs to train the graduates and provide them strong research infrastructure to hone their skills for improved quality human resource that don’t migrate and supports the local ecosystem [89]. Although Indian R&D has produced a sizable scientific workforce, very few scientific leaders and very few ideas have achieved commercial success. To create a high-value knowledge economy, universities should encourage interdisciplinary research. An Indian researcher typically comes up with solutions to research problems without considering any potential applications for industry. Success depends on carefully considering every step of the innovation’s journey from the research lab to the patient’s bedside. To ‘close the loop’, the way such challenges are approached must alter, and study findings must consider their commercial viability. Universities and research institutions should employ personnel with expertise in intellectual property (IP) and technology transfer. Academicians and researchers should work together with industry to license their inventions and promote the results of their study. A more robust framework of industry–academia partnership might lead to the development of technological commercialization techniques, which would advance social development and the economy. The redesign of a research-based curriculum emphasizing the enhancement of a domain-specific skill set, more vocational hands-on trainings, improved publishing practices, faculty enrichment programmes, student exchange programmes, merit-based incentives, and awards for researchers and promoting research institute, university, and industry collaboration are just a few Indian research areas that need improvement to match the global arena [90].

### 7.7. Status of Industry-Academia Partnering

Although India boasts many universities, research centers, and a relatively skilled workforce, the global university–industry collaboration ranking of India is 24, the IPR ranking is 47, and innovation ranking is 39. This reveals a seemingly wide industry–academia gap that needs to be bridged. Many successful research findings in academia, even though promising, may not translate to novel commercialized technologies or patents. They are not getting the opportunities due to a lack of industrial collaborations, lack of IP knowledge, and due to the practices of multiple publications in stringent timelines [91,92,93,94,95,96]. 

India may be inspired by Israel biotechnology momentum [97]. The Israeli government supports many programmes that have improved the biotechnology sector through innovation and skill development in biotechnology in general and medical research. The funding for life science research is half of its total research funding. The government supports start-ups after a proof of feasibility phase to success, which is crucial for the start-up’s survival. Israel’s high-tech incubators are public-private partnership ventures, nurturing young biotech companies by offering R&D facilities, experienced management, as well as financial and administrative support. The Magnet framework, a consortium of industries and research institutions to develop innovative technology, was established by the Israeli government for this. Academic research groups are engaged in scientific or technological research to promote applied research and commercialize the technologies as per industry need. Then, manufacturing companies develop competitive and innovative products. ‘Magnet’ also supports high-tech incubators providing a home for innovative project development. It provides long-term financial support with exemption of royalties to industries, promoting a solid framework for ground-breaking innovations. Bioplan 2000 supports biotech incubators through funding, providing infrastructure and management. Along with this, the Ministry of Science, Culture and Sport has supported biotechnology as ‘national programme’, where various research groups were involved in developing the skills and improving infrastructure and fund allocation for academic biotechnology and medical research. Israel’s leading biotech companies are built on academic excellence, an enabled workforce, entrepreneurial endeavor, high-end technologies, extraordinary funding support, and skilled management [98].

Indian government slowly though surely has been taking several steps to build an industry-academia collaboration ecosystem. A dedicated ‘Entrepreneurship and Skill Development’ Ministry has been recently established to promote young professionals for entrepreneurship and to train the manpower as per the industry need. Atal innovation mission intends to establish Technology Business Incubators (TBIs) in universities. The National Initiative for Developing and Harnessing Innovations (NIDHI), Promoting Innovations in Individuals, Start-ups and MSMEs (PRISM), Impacting Research, Innovation and Technology (IMPRINT) are few other programmes for industry–academia collaboration. The National Biopharma Mission ‘Innovate in India’ is a mission for industry–academia collaborations established by Department of Biotechnology (DBT) in collaboration with the World Bank. The program is devoted to technological and product development in the biopharmaceutical sector to enable stakeholders to become globally competitive. This specifically focused on the development of vaccine, biosimilars, therapeutics, and diagnostics. Additionally, there are several fellowships, e.g., the Prime Minister Fellowship, Department of Science and Technology (DST), and CSIR-Industry sponsored research scheme initiative, from the government to strengthen innovation and contribute to national economy through industry–academia links [99,100].

### 7.8. Mindset Issues—Need to Embrace the State-of-Art Technologies

Biopharma-related legal experts opine that although India’s domestic biosimilar market is rising, its international business may be impeded due to a loose regulatory structure that makes other nations wary of the quality of the biosimilars. There are 98 approved biosimilars in India, with at least 50 on the market—the most of any country. In comparison, according to a WHO survey, the US has 26 approved biosimilars and the European Union has 61 [101]. The heart of the operation is bioprocess technology that has developed significantly in last decades. State-of-the-art techno-management approaches, e.g., quality by design (QbD), process analytical technology (PAT), single use technology, just-in-time, or lean manufacturing, are becoming common platforms in the biopharma sector worldwide. In contrast, Indian biopharma developing biosimilars still rely hard on older technologies and remain to adopt these new norms, allegedly citing justifications, such as the lack of relevance or the cost pressure. A holistic view of project planning and risk assessment in terms of resources and timeline is critical to manage cost pressure. The development and implementation of balanced a quality management system (QMS) will reduce the cost of production and protect the data through an achievable and retrievable system. It also controls equipment and process management. Use of QMS can greatly manage the time for process development and application for approval process. Although adopting ‘single use’ has gone up in Indian biopharma recently, stainless steel systems in manufacturing still dominate. Moreover, many are yet to deploy sophisticated analytical tools [102].

### 7.9. Lack of Internal (Industry-Exposed) Expertise

In terms of having the right knowledge ecosystem and pool of talent, India is critically lagging. The Indian education system is still very theoretical and hardly exposes graduates and post-graduates to a high level of practical hands-on experience, particularly in the biopharma sector [103,104,105,106]. Most universities and academic institutions in India are not research oriented and highly theoretical. The curriculum designed by the institute is based on theoretical approaches and only the related practical programmes. However, these curricula do not encourage including the factual issues and possible solutions through life science and practical problem-solving approaches. Many business leaders and senior managers in the Indian biotech sector emerged from the run-of-the-mill pharma base and struggle to fully understand and cope with the fine nuances of new-age biotech drug development. This results in misaligned processes and analytics not meeting the regulatory expectations. Obviously, the regulators will be eagle-eyed on quality that would seemingly pose a challenge to Indian companies. There is hardly any curriculum covering quality culture or taught in any Indian university or higher education program, exposing young professionals to a global regulatory framework and quality culture. It has resulted in a few drugs being barred in key overseas markets in recent years, a distraction for Indian biopharma manufacturing hubs. The hard reality is that many Indian biotech drug makers are still struggling to fix such regulatory issues in quality manufacturing operations. The companies should design and timely assess need-based training programmes. Such training will create the opportunity to develop skill sets related to industry and job-specific techniques, e.g., leadership, management, general business, manufacturing, finance, and the overall techno-management. Biocon academy is a trendsetter in an industry-ready workforce, offering interdisciplinary courses, helping professionals and technicians grow through industry–academia interaction.

### 7.10. Budget-Funding

In developed markets, e.g., the US [107], UK [108], EU [109], etc., most of the biotech innovation is fueled by small-time emerging bio-ventures primarily funded by angel investors or venture capitalists. Through alliances, collaborations, or acquisitions, new technological platforms or discoveries eventually find their way to significant biotech companies. In India, however, the lack of new and small biotech companies with creative platforms is more pronounced. Investors won’t put their faith in the little Indian biotech start-ups because of their perception of them as cheap global manufacturing hubs or as companies creating low-cost knockoffs of popular medicines. An overly ambitious company plan, a lack of a focused execution strategy, inadequate risk management, and a lack of planning ahead of time all contribute to start-up failure. Lack of significant, effective industry–academia relationships and a suitable entrepreneurial ecosystem to support biotech start-ups are detrimental and eventually lead to failure in capturing a suitable market with a particular client base [110,111,112].

### 7.11. Market Needs and Response to Competitive Pressures

With a thriving domestic biosimilars market, India ranks first in the number of approvals (98), but the country’s guidelines for biologics development are not considered to be as effective as those by the US and the EU, or the WHO in general. To address the issues and challenges associated with developing biosimilars, the Central Drugs Standard Control Organization (CDSCO) in collaboration with the Department of Biotechnology (DBT) developed ‘Guidelines on Similar Biologics: regulatory requirements for marketing authorization in India’ in 2012 and revised it in 2016. It endeavors to align it with international agencies like EMA, USFDA and the WHO [113].

Notwithstanding the domestic regulatory framework, a few of the mandatory global aspects for the biosimilars regulatory requirements immediately applicable to India are:
1.Interchangeability: FDA needs ‘an interchangeable biological product which is similar to an existing FDA-approved reference product’. This allows substitution of the reference product with the interchangeable biologic by a pharmacist without the interference of the clinician who prescribed the reference biologic [114].2.Naming: WHO follows the International Non-proprietary Name (INN) for generic products. Several other countries have adopted their unique naming convention. EU follows INN while Japan adopt INN with BS as suffix, US also follows INN with four-letter suffixes [115].3.Labelling: After approval, the insert should clearly indicate whether the data were generated on a similar biologic or innovator product, including differences in characterization and the extent of similarity with the reference biologic on safety, immunogenicity, and efficacy for the awareness of patient and professionals. Moreover, the COOL (country of origin labelling) law applies, since 2003 [116].

### 7.12. Filling the Gaps in International B2B and B2C Collaborations, and Handholding

Despite decades of dominance by the generic medicine sector, India’s participation in the race to produce complex biotech medications, a worldwide market worth tens of billions of dollars, is quite pitiful. While there are few such items available on the local market in India, where regulatory barriers are relatively low, South Korean, American, and European companies are quickly catching up in the race to supply the lucrative Western markets. Only three Indian companies, Biocon, Dr. Reddy’s Laboratories [117], and Intas Pharmaceuticals, have partnerships that are effective in developing biosimilars for the Western (US and EU) markets [118].

For many smaller Indian players, the expense and complexity of creating biosimilars have acted as major barriers. The three Indian start-ups that have announced aspirations to produce biosimilars for the US and Europe have all teamed up with more established Western enterprises. Through a partnership with Mylan Inc., four compounds are being tested by Biocon in Phase III trials. The testing criteria in India do not adhere to the USFDA, EMA, and WHO norms, which many nations with biosimilar markets either adopt or model through their own national guidelines. This is a major problem for India’s attempts in international biosimilar trade. Indian biosimilars must pass the tests in accordance with what is generally regarded as scientifically sound for them to be taken seriously by international regulators. Ideally, a worldwide uniform approach to biosimilar approval would consider the current regional variations in rules and ensure that the ‘head-to-head’ similarity concept at the center of strong guidelines from the EMA, FDA, and WHO is retained [119,120].

Historically, many Indian biosimilars had to revisit their product development strategy to stand a chance in the US market. Many believe that India’s technical expertise, vast experience with the generics’ development, and the huge density of scientists will help them overcome international challenges. Indian biosimilars market remains robust, as evidenced by the large number of biosimilars, and there have been international successes in a few cases too. Biocon is one such example of an Indian biopharmaceutical company successfully entering the US market [121].

### 7.13. Lack of Diversification

The present Indian biotech industry has its roots in the traditional generic pharmaceutical industry. The $15 billion pharmaceutical sector in India has long been based on copying chemical medications. However, because biotech medications are more challenging to produce and duplicate, authorities have developed the idea of equivalent versions that are functionally equivalent. Though biotech medications that need genetic engineering make up an increasing portion of innovative medications, the outlook for copycat medicines is dismal and they will move into the straightforward small-molecule pharmaceutical category. As was previously said, Indian biopharma is hampered by a lack of trained and qualified personnel. The Indian pharmaceutical industry should make investments in training and human resource development to maintain the effectiveness of its talent pool. The personnel should receive ongoing training that covers technical topics as well as current regulatory rules, international standards, and patent laws, among other things. Late bloomers, such as China [122,123,124] and Korea, moved far more quickly by partnering with significant international firms of the highly regulated international market than India, who joined the biosimilars market first. Most manufacturing facilities adhere to cGMP and are regularly accredited by the FDA and EMA, giving them access to international collaboration. The regulatory investment opens the platform for multinational trials, and they are implementing policies similar to those used on a global scale to improve the quality and credibility of clinical trials. In contrast, the Indian biopharma sector is having problems due to a lack of policy support and pricing control. They are mostly moving toward biobetters (better safety and effectiveness profiles, enhanced ADME-Tox profiles, less side-effects, increased functionality, longer stability, better formulation, etc.) or cutting-edge biotech medications throughout time. Sadly, India still places a lot of emphasis on the traditional biosimilars, particularly those that are losing their patents. This leads to unhealthy competition and little to no distinction. Before biosimilars were introduced to Europe in 2006, they were already available in India in the early 2000s. The first biosimilar was introduced on the market in the US following the recent introduction of a regulatory guideline [125,126,127].

India’s experience has not been good. Intas, for instance, recently received reports of patients on its biosimilar version of Roche’s eye drug Lucentis developing inflammation barely two months after the drug was launched [128]. The CDSCO and DBT guidelines have enabled manufacturers to bypass Phase III clinical trials in circumstances where sufficient pharmacodynamic (PD) and pharmacokinetic (PK) data are available, opening the floodgate for faster product approval. As such, drug developers in many emerging countries including India face heightened scrutiny related to data integrity breaches. Such a plan to bypass phase III for biosimilars could be a serious global concern in times to come. It can raise questions on the GMP of related manufacturing companies, difficulty in penetrating global markets, and create concern among patients and physicians on the quality and safety of the biosimilar.

Having a system in place that defines what tests need to be done is critically important. Guidelines have relieved Indian manufacturers of conducting redundant studies, making the process of bringing a product to market more feasible. Prior to the revisions, a reference product needs approval and marketed in India if the manufacturer desired its biosimilar to treat as that of a reference product. The alterations allow Indian companies to develop biosimilars based on reference products approved by the US, European Union, Japan, Canada, and Switzerland [129]. Additionally, CDSCO and DBT included a requirement for Phase IV trial post marketing, which must include at least 200 trial volunteers and be conducted within two years of market approval [130].

Most Indian companies engaged in biosimilars manufacturing do not attempt to crack the US market. There seems to be eagerness from Europe and South America to develop and expand biosimilars in the US. The manufacturing process of quality biosimilars is complex. The regulatory frameworks in US are stringent, having broad IP and patent laws, which may lead to a major obstacle in effective marketing leading to patient inaccessibility and cost-increase. Different pharmaceutical companies are already in the global market, providing tough competition. The cost of litigation, if any, is quite a high burden on Indian companies which forces them to shy away. Biocon has been willing to battle out with patent-holders in court. In March 2020, for instance, the company won a court-ruling that invalidated a Sanofi patent which attempted to block Biocon and Mylan from commercializing an insulin product, Glargine, in the US [131,132].

### 7.14. Regulatory Affairs and IPR

Indian regulations stem from biosimilar guidelines originally drafted in 2012 (revised in 2016) by two Indian agencies, the Central Drugs Standard Control Organisation (CDSCO) and the Department of Biotechnology (DBT). There are conflicting opinions on whether the revision of the guidelines has made a difference in ensuring the ‘inspired’ biosimilars perform similar or better than their originator (inspirer/reference) products.

The patent expiry of many largely successful biologicals has paved the way to develop biosimilars as alternative to the otherwise high-cost biological therapies [133,134]. Given the complexity of biologicals, the regulatory guidelines for biosimilars approval are meticulous and different from the generics. Hence, biosimilar developers often face issues in applying for evaluation by regulatory authorities. With large number of biosimilars at the development stage, a deeper understanding of the regulatory approval process by the manufacturer is needed. A major deficiency identified is the comparability between the proposed biosimilar and its reference biological. As stated earlier, there are conflicting opinions on differentiating between the ‘inspired’ biosimilar and the ‘inspirer’ product. Regulatory bodies recommend a three-tier comparability approach to gain regulatory approval for biosimilars, viz., analytical characterization, preclinical trials, and clinical studies. Many applications from India stall due to a failed pharmacokinetic (PK) comparability study. PK studies are considered more sensitive to detect product-related differences. Process validation is another major issue, wherein the common deficiencies observed relate to the process controls, validation of equipment sterilization, vials-filling strategy, profiling impurities, and the stability of the bioactive substance/product.

It is important for biosimilar manufacturers to understand the reference molecule via an in-depth analysis and identify critical quality attributes (CQAs) that may impact the safety and efficacy of the biosimilar. The manufacturer needs to design a process and establish appropriate process controls to ensure product reproducibility and process repeatability with consistent product quality. Stability studies are critical to demonstrate the shelf-life and sustained quality of products. To develop biosimilars, the manufacturer must follow good manufacturing practises (cGMP) and good clinical practises (GCP) for adequate cGMP and GCP compliance as may be desired by the regulators [135,136,137].

Regulatory experience of biosimilar manufacturers reveal that most rejections were due to the gaps in the manufacturer’s understanding of regulatory expectations from a comparability exercise, and cGMP and GCP compliance. Manufacturers must learn from experience with time, leading to fewer rejections. Although Indian guidelines are revised, they still fall short on various counts, including animal testing. For instance, the guidelines suggest that monoclonal antibodies (mAbs) be tested on rats, which will obviously provide limited valuable data on the efficacy of the drug on humans. Immunogenicity tests to measure the potency of a biological agent to stimulate an immune system response are conducted solely in animals, which fails to provide a full picture of how biosimilars will influence humans, owing to dissimilar metabolic responses [138,139,140]. mAbs have no toxicity in rodents as they do not possess the complimenting receptors. The efficacy and safety testing can be performed in 3D cell culture and xenograft model which often exhibit more similarity to in vivo tissue organs regarding gene/protein expression profiles [141]. The CDSCO in 2016 had established guidelines based on WHO specification. mAb characterizations are also consistent with the WHO framework. It defines in vitro cell-based assays for cell proliferation, cytotoxicity, receptor binding, and neutralizing assay but safety and in vivo PK/PD criteria are still ill-defined. The biological assays are determined, but the types of assays are not specified. Biosimilars must be similar to the reference products in terms of safety and efficacy, so the needed tests and their results are critical. Drugs should not significantly exceed the efficacy of the reference product they are intended to imitate, as a biosimilar with greater potency than the reference product may have more serious latent adverse effects. Improper standards may mean some products do not truly qualify as biosimilars; the drug being effective does not prove to be biosimilar. Moreover, as many Indian trials recruit fewer (100 or so) volunteer participants, the statistical validity of such trials is questionable.

Unlike in the US, India does not require biosimilars to pass an ‘interchangeability’ test, allowing the pharmacists to switch patients from reference products without physician permission. The US is the only nation requiring a biosimilar interchangeability designation. This designation limits patients’ access to cheaper biologics. Another big difference is the perception of biosimilars. Whereas the US physicians and patients may be uninformed or misinformed about the safety and efficacy of biosimilars, the Indian government encourages people to search for and adopt cheaper alternatives. Statistics suggest that the Indian population is more price-sensitive on healthcare. Many households do not have health coverage, and 82% of healthcare costs in India are paid out of pocket. A cheaper alternative for a prescribed drug, therefore, will naturally hit the market. Another difference in views between India and the US is the rate of uptake of biosimilars. This rate is rapidly growing in India, and a bigger reason for this is the high cost to private investors in pharmaceuticals research and development, making the advanced drug too costly for the commoners’ reach [142,143,144,145].

## 8. Indian Vaccines in the Global Market

India has experienced a momentous journey from being an importer to the largest global exporter of vaccines of immediate and long-term healthcare significance. India contributes nearly 50% of the global vaccine demand for immunization programmes. King Institute, Guindy in Tamilnadu was established as a BCG lab immediately after Indian independence, in 1948. Subsequently, India led in the production and export of six vaccines (BCG, TT, DPT, DT, polio, and typhoid) for children under the expanded immunization programme, and Measles vaccine was added to the list later in 1985 [146]. Similar to the global scenario until the 1980s, vaccine requirements in India too were primarily fulfilled through government institutions. This could be attributed to the fact that the Indian public sector lacked an access to novel technologies to produce vaccines for TT, DT or DTP, or the scale-up production of oral polio or the measles vaccines. In 1987, the Bharat Immunologicals and Biologicals Corporation Ltd. (Bulandshar, India) was established by the Department of Biotechnology, Government of India in technological collaboration with the Institute of Poliomyelitis and Viral Encephalitis (Moscow, Russia). The production was initially restricted to the repackaging of OPV that was imported in bulk from the Russia [147]. The second phase targeted at indigenous OPV production in the next five years. The Indian Vaccine Corporation Ltd. (IVCOL), Gurgaon, Haryana initiated indigenous measles vaccine production in technological collaboration with Institut Merieux, Lyons, France. However, it could not peak further as the private sector that acquired the Institut Merieux denied transfer of technology to IVCOL. IVCOL was shut down, and India imported measles vaccine to fulfil its requirement until the Serum Institute of India (SII), Pune started its supply to the Expanded Programme on Immunisation (EPI) EPI (now called Universal Immunisation Programme (UIP) in India) in 1992 [147].

The Hepatitis B vaccine launched in 1990 became the game changer [148]. The Indian private sector took the driver’s seat gradually and launched several vaccines, e.g., influenza, MMR, and chickenpox, that found their way into regular global immunization projects. Since then, driven by the increased investments and favorable government policies, the Indian private sector vaccine market revenue reached US $95 billion in 2020. As it stands now, the Serum Institute of India Pune, Indian Immunological Limited (a subsidiary of National Dairy Development Board), BCG Chennai, BIBCOL Uttar Pradesh, Wockordt Limited Mumbai, Shanta Biotech (Sanofi) Hyderabad, Zydus Cadilla Pharma Ahmadabad, Bharat Biotech, Biological E, Panacea Biotech, etc. represent the major vaccine producers [147]. Indian vaccine industries, especially Serum Institute and Bharat Biotech Ltd., played a significant role in the global vaccination drive during the COVID-19 pandemic (Table 2 and Table 5) [149]. Amid the rising Monkeypox (MPX) cases in India, the Serum Institute of India (SII) is set to manufacture a vaccine against it [150]. The National Institute of Virology (NIV, Pune) could successfully isolate MPX virus from a patient sample to help develop vaccine and test kits (https://www.livemint.com/news/india/monkeypox-8-cases-in-india-so-far-1-death-10-things-to-know-11659430193160.html, 8 October 2022). Vaccine manufacturers and vendors need to collaborate with the healthcare systems and researchers to earmark the global need for the production scale-up. Collective regional and global partnerships are highly imminent to strengthen and execute the action plans.

## 9. Visionary Perspective and Conclusion

Around 20% of generic medications and 62% of vaccines are now supplied by the Indian pharmaceutical industry, which has emerged as the world’s leading supplier of pharmaceuticals (by volume). A very positive sign was the remarkable work done during the most recent COVID-19 pandemic. India’s biopharmaceutical industry has been instrumental in producing and supplying treatments to fight COVID-19 since the first days of 2020. Industries increased their output to provide more than 133 nations with high-quality, reasonably priced medications despite a weak supply chain and a lack of labor. Six Indian pharmaceutical companies are currently producing the antiviral medicine “Remdesivir” due to strong demand and distribution in 127 countries to combat the continuing pandemic. In India, 30 groups from the pharmaceutical industry and academia are working on the COVID-19 vaccine, and most of them have received preliminary approval for use in an emergency. India is making good progress with its immunization campaign. As previously described, there are gaps in effective translational research on Indian soil that must be promptly and appropriately filled. The adoption of cGMP in various manufacturing firms with accreditation from international regulatory bodies and the improvement of regulatory rules on a worldwide scale will encourage Indian producers of biosimilars to compete on the global market [151]. Innovations, working with other sectors of the economy, and promoting training programmes to keep a strong scientific workforce will help to advance to world-class potential. The government has several programmes for the start-ups for small and medium biotech enterprises, industry-academia collaborations, and translational research. Such inventiveness would help the Indian biopharma sector to improve its global competitiveness and reach the target of a US $5 trillion economy by 2025. The two major areas for governmental support and interventions are the state-of-art technological infrastructure and the high-end industry-ready trained manpower. These two should follow a proactive and encouraging handholding in terms of administrative and regulatory frameworks in the spirit of ‘ease of doing business’. The time is ripe to grow our economy through evolution in the biotechnology and biopharma sector. Innovation demands a strong and effective collaboration among university, industries, and incubators. Public-private partnership should nurture more start-ups and take steps for skill development, where we are lagging. India needs to learn the investment strategy, development of innovative and managerial skills, adoption of new techniques, collaborations, and business models from countries, such as Israel, South Korea, Japan [152,153], China [154], and USA. Indian society and the mindset of its people should be supportive towards a risk-taking attitude and encouraging to innovation driven changes. By empowering technopreneurs to develop from small and medium enterprises to large industrial scale operations, India will be able to take innovative thoughts to the global level and ultimately realize an ‘*Atmanirbhar Bharat*’ (self-reliant India).

With a detailed and closely researched compilation of the state-of-art facts and figures, this article is an effort to provide a 360° view of how Indian biosimilars and vaccines could replicate the success of pharmagenerics at the global stage. However, as is universally accepted for scientific works, there are few limitations as well. They are pointed out below. Firstly, the details about the performance of the upcoming and greenhorn (start-up) companies in this greenfield sector are lacking and rather unorganized wherever they are available. Secondly, there are few such new-age biosimilar products which are quite difficult to differentiate from the biologics, thus making the picture on the financial projections and global demand trends for biosimilars alone hazy. Thirdly, being a greenfield sector, the sectoral dynamics remains ill-understood. Fourthly, the majority of the potential biosimilars and vaccines products are at various stages of their development in India and elsewhere, and their ultimate impact on the overall global biosimilars scenario will be clear in due course of time. Finally, India replicating its success in pharmagenerics in the innovation and research-intensive field of biosimilars and vaccines at the global stage would have to face strong challenges from countries, such as the US, China, Korea, Japan, and the European Union, which will be revealed better with the passage of time.

## Figures and Tables

**Figure 1 vaccines-11-00110-f001:**
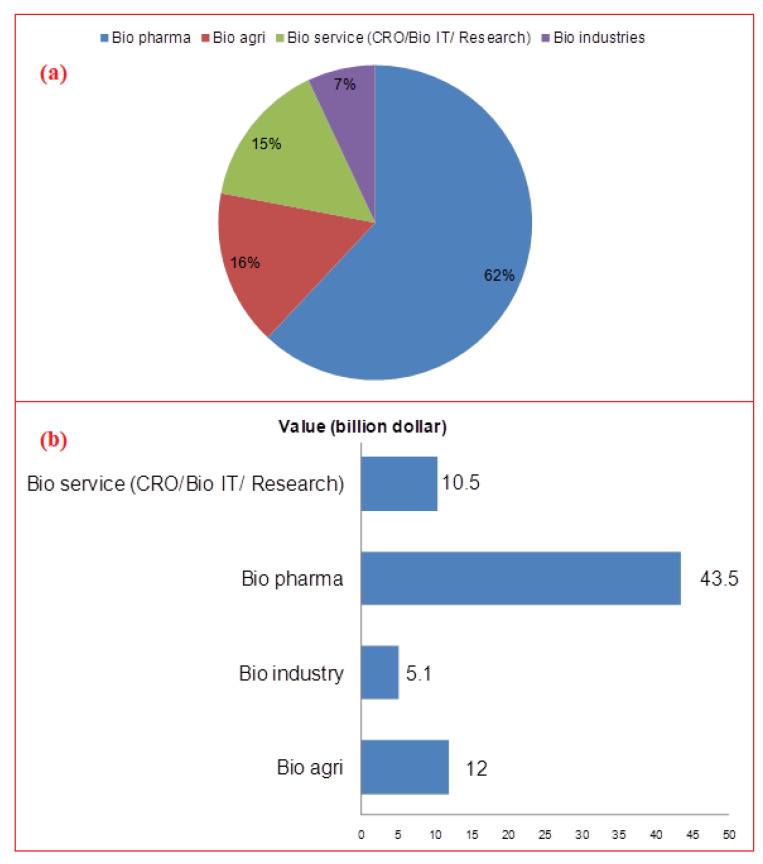
(**a**) Percent share of various biotechnology sector segments in India; (**b**) Contribution of different biotech sector segments towards Indian economy at 70.2 billion dollars in 2020 [13].

**Figure 2 vaccines-11-00110-f002:**
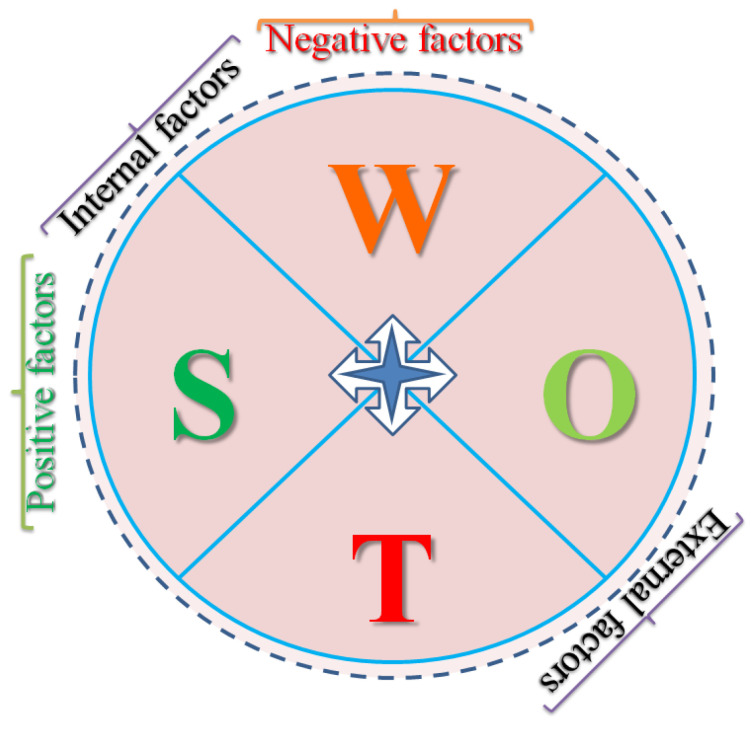
The SWOT chart depicting the interrelationship and correlation of growth parameters that affect a corporate ecosystem.

**Table 1 vaccines-11-00110-t001:** The organic growth scenario of Indian bioindustries.

Triggering Factor	Beneficiary Organisation	Speciality Area	References
Scientists or local individual from various industrial sectors	XCyton Diagnostics (Bengaluru)	Diagnostics	[21]
	GangaGen (Bengaluru)	Antimicrobials	
	Shanta Biotechnics (Hyderabad)–now part of Sanofi	Biogenerics, diagnostics, contract research	
	Cytogenomics (Bengaluru)	Bioinformatics	
	Bigtec (Bengaluru) Brilliant Bio Pharma Private Limited (Hyderabad)	Bioinformatics Veterinary Vaccines	
Companies venturing into biotech	Serum Institute of India (Pune)	Vaccines, biosimilars	[21]
	Biocon (Bengaluru)	Generics, biologics, biosimilars	
	Infosys (Bengaluru); Tata Consultancy Services (Mumbai)	Bioinformatics	
	Zydus Cadila	Generics, biologics, biosimilars	
	Bioogical E (Hyderabad)	Vaccines. Biologics	
	Intas Pharmaceuticals (Ahmedabad)	Generics, biogenerics, contract manufacturing	
	Emcure–Gennova (Pune)	Biosimilars, Novel Vaccines	
	Panacea Biotec (New Delhi)	Generics, vaccines	
	Wockhardt (Mumbai)	Generics, biologics, vaccines	
	Dr. Reddy’s Laboratories (Hyderabad)	Generics, vaccines, biosimilars, biologics	
	GVK Biosciences (Hyderabad)	Generics, biogenerics	
	Jubilant Biosys (Bengaluru)	Bioinformatics, contract research	
Academic scientist to bioentrepreneur	Bangalore Genei (Bengaluru)	Reagents supply, contract research	[21]
	Avesthagen (Bengaluru)	Plant biotech, diagnostics, nutraceuticals, contract research	
	Strand Life Sciences (Bengaluru)	Bioinformatics	
	Microtest Innovations (Bengaluru)	Diagnostics	
Industry professionals or academic scientists from overseas	Molecular Connections (Bengaluru); Cell	Bioinformatics, systems biology	[12,21]
	Bhat Biotech (Bengaluru)	Diagnostics	
	Bharat Biotech International (Hyderabad)	Vaccines, Biosimilars	
	Genotypic Technologies (Bengaluru)	Bioinformatics, contract research	
	Connexios Life Sciences (Bengaluru)	Systems biology	
	Ocium Biosolutions, Mapmygenome (Hyderabad)	Bioinformatics, Diagnostics	
Multinational company setting up in India	Quintiles India (Bengaluru); Accelrys (Bengaluru)	Bioinformatics, contract research	[21]
	Merck KGaA (Bengaluru); MWG (Bengaluru); Thermo-Fisher (Bengaluru); Sartorius (Bengaluru); DuPont (Hyderabad)	Reagents and equipment supply, customised bioservices; R&D facilities	
	GlaxoSmithKline Pharmaceuticals Ltd., Mumbai	Generics, Vaccines	
Govt. initiated biotech company	Indian Immunologicals (Hyderabad); Human Biological Institute (Ooty)	Vaccines (Animal and Human)	-
	Bharat Immunological and Biologicals Corporation Limited (BIBCOL), (Bulandsahar, UP)	Vaccines	
	Indian Vaccine Corporation Limited (Delhi)		
Start-up as emerging biotech company	Med Genome (Bengaluru), X Code Life (Chennai), FARCAST Biosciences (Bengaluru)	Bioinformatics, Diagnostics, drug dicovery	-
	BUGWORKS (Bengaluru)	Antimicrobials	
	Pandorum (Bengaluru)	Tissue Engineering, Regenerative medicines	
	Oncostem Diagnostics (Bengaluru)	Diagnostics and therapy	
	Zumutor Biologics (Bengaluru)	Novel Immunotherapy and stem cell research.	

**Table 2 vaccines-11-00110-t002:** Dominant biotech companies in global biosimilars market with market share of the US and others, except India.

Industry	Country	Flagship Biosimilar	Market Share
Johnson and Johnson	USA	Remicade (Infliximab) [31]	48.2%
Pfizer	USA	Inflectra^®®®^ (infliximab-dyyb in the US) [32]	
Mylan	USA	Ogivri (Trastuzumab) [33]	
Biogen	USA	Byooviz [34]	
Eli Lilly	USA	Insulin Glargine [35]	
Coherus Bioscience	USA	Cimerli (Ranibizumab-eqrn) [36,37,38]	
MSD (Merck & Co)	Germany	Ontruzant (Trastuzumab) [39]	18.1%
Boehringer Ingelheim	Germany	Cyltezo (adalimumab-adbm) [40]	
Fresenius Kabi AG	Germany	Stimufend [41]	
StadaArzneimittel AG	Germany	Silapo (epoetin-zeta) [42]	
mABxience	Switzerland	Novex (rituximab) [43]	
Roche	Switzerland	Lucentis (ranibizumab) [44]	
Sanofi Aventis	France	Lovenox (enoxaparin sodium) [45]	
GlaxoSmithKline	British	Nucala (mepolizumab) [46]	
Teva Pharmaceutical	Israel	Truxima (rituximab-abbs) [47]	
Gan and Lee Pharmaceuticals	China	Glargine [48]	
Amega Biotech	Argentina	Neutropine (Filgrastim) [49]	
Samsung Biologics	South Korea	Byooviz [34]	
Celltrion	South Korea	Remsima [50]	
Chong Kun Dang	South Korea	Darbepoetin Alfa [51]	
Probiomed	Mexico	Trastuzumab [52,53]	
Apotex	Canada	Apobiologix (pegfilgrastim) [54]	
JCR Pharmaceuticals	Japan	Agalsidase beta [55]	
Gedeon Richter	Hungary	Terrosa [56]	
Biocad	Russia	Trastuzumab [57]	

**Table 3 vaccines-11-00110-t003:** A list of the approved Indian biosimilars which accounts for only 3% of global share.

Product	Company Name	Active Drug Molecule	Therapeutic Use in
Glaritus	Wockhardt	Insulin Glargine	Diabetes
Grafeel	Dr. Reddy’s Laboratories	Filgrastin	Neutropenia
Pegfilgrastism	Lupin	Pegfilgrastin	Cancer, Neutropenia
Epofer	Emcure	Epoetin alpha	Anemia
Zyrop	Cadila Healthcare	Erythropeotin	Chronic kidney failure
Krabeva	Biocon	Bevacizumab	Colorectal cancer
Bevacirel	Reliance Life Sciences		
Cizumab	Hetero		
Erbitux		Cetuximab	Colorectal cancer
Acellbia	Biocad	Rituximab	NonHodgkin Lymphoma
Maball	Hetero Group		
maTabs	Intas Pharmaceuticals		
Adafrar	Torrent Pharmaceuticals	Adalimumab	Rheumatoid Athritis, Crohn’s disease
CaNMab	Biocon	Transtuzumab	Breast cancer
Intacept	Intas Pharmaceuticals	Entanercept	Rheumatoid Athritis
Relibeta	Reliance Life Sciences	Interferon Beta 1a	Multiple sclerosis
Razumab	Intas Pharmaceuticals	Ranibizumab	Degenerative myopia
AbcixiRel	Reliance Life Sciences	Abciximab	Angina, Cardiac ischemia
Basalog	Biocon	insulin glargine	Diabetes
Biovac-B	Wockhardt	hepatitis B vaccine	Hepatitis B
FostiRel	Reliance Life Sciences	follitropin beta	Female infertility
Mirel	Reliance Life Sciences	reteplase	Myocardial Infraction
Zavinex	Cadila Health Care	Interferon alfa-2b	Chronic hepatitis B and C
Choriorel	Reliance Life Sciences	chorionic gonadotrophin hormone r-hCG	Female infertility

**Table 4 vaccines-11-00110-t004:** List of some Indian government, public sector undertaking and private sector vaccine manufacturers.

Sl. No.	Manufacturer	Licensed Vaccine	Target Species	Reference
1.	BCG Vaccine Laboratory, Guindy, Tamilnadu, India.	Tuberculine, BCG	Human	[63,64]
2.	Pasteur Institute of India, Coonoor, The Nilgiris, Tamilnadu, India.	DTP, DT, TT and inactivated rabies vaccine	Humans and canine	[64,65,66]
3.	Central Research Institute, Kasauli, Solan, Himachal Pradesh, India.	Yellow fever, JE, DTP, DT, TT	Humans	[64,67]
4.	BIBCOL, Chola, Uttar Pradesh, India.	bOPV	Human	[64,68]
5.	Haffkine, Parle, Mumbai, India.	bOPVand mOPV	Human	[68]
6.	Human biological Institute, a division of Indian Immunologicals Limited, Hyderabad, Telangana, India.	Rabies, DTP, TT, DT, Hep- B, Pentavalent (DTP+Hib+HepB)	Human and canine	[64,68]
7.	HLL Biotech Ltd., Taramani, Chennai, Tamil Nadu, India.	Hep B, DTwP- HepB-Hib	Human	[68]
8.	Bharat biotech International Ltd., Hyderabad, Telangana, India.	Hib, Rabies, bOPV, mOPV, DTP+Hib+HepB, Vi polysaccharide Typhoid, H1N1, DTP, DTP+HepB, Rotavirus vaccine, Inactivated JE vaccine, Typhoid+TT Conjugate Vaccine andDTP+Hep- B+Hib (Liquid), DTP+Hib, BBV152 Covaxin	Human, canine	[64,68]
9.	Biological E, Hyderabad, Andhra Pradesh, India.	DTP, TT, JE bulk & DT	Humans	[64,67]
10.	Biomed Pvt. Ltd., Ghaziabad, Uttar Pradesh, India.	Hib, Meningococcal Polysaccharide, bOPV, Rabies, Meningococcal polysaccharide.		
11.	Cadila Healthcare, Ahmedabad, Gujarat, India.	Rabies, H1N1, trivalent influenza	Human	[64,68]
12.	Serum Institute of India, Pune, Maharastra, India.	DTP, TT, DT, Hep-B, Hib, MMR, Measles, Rubella, BCG, IPV, DTP+HepB+Hib (Liquid+lyophilised), DTP+HepB, DTP+Hib, H1N1, Meningococcal A conjugate (lyophilised), Mumps, MR, H1N1(whole virion inactivated), Measles+Mumps, Measles+Rubela, Seasonal Influenza vaccine, COVID-19 vaccine	Human	[64,67,68]
13.	Shantha Biotechnics Ltd., Hyderabad, India.	DTP, DTP+HepB+Hib (Liquid), DTP+Hib, DPT+Hep B, TT, Hib, Hep-B, DT bulk, TT Bulk, Hib Bulk, Hep B Bulk, DTP bulk, DTP+HepB+Hib bulk, DTP+HepB+Hib RTF bulk, Oral cholera vaccine, IPV RTF Bulk, IPV	Human	[64,67,68]

**Table 5 vaccines-11-00110-t005:** Various Indian biopharmaceutical companies engaged in COVID-19 vaccines development and manufacturing (https://covid19.trackvaccines.org/country/india/, 9 October 2022).

Vaccines	Indian Manufacturer	Collaborator(s)	Current Regulatory Status
Covishield	Serum Institute of India, Pune, India	Oxford-AstraZeneca	Approved
Covaxin	BharatBiotech Int. Ltd., Hyderabad, India	Indian Council of Medical Research, National Institute of Virology	Approved
ZyCov-D	Cadila Healthcare Ltd., Ahmedabad, India	Department of Biotechnology, India	Approved
Sputinik V	Dr. Reddy’s lab, Hyderabad, India	Gamaleya National Centre, Russia	Approved
NCV-COV2373	Serum Institute of India, Pune	Novovax	Emergency authorisation
HGCO 19 m-RNA based vaccine	Genova, Pune, India	HDT-Bio, US DBT	Approved
Recombinant protein-based Vaccine (Corbevax)	Biological E, Hyderabad, India	Baylor College, US	Approved
Codon-deoptimised live attenuated COVID-19 Vaccine	Indian Immunologicals Limited, India	Griffith University, Australia	Pre-clinical
Warm COVID-19 Vaccine	Mynvax, Indian institute of Science, Bengaluru, India	BIRAC	

**Table 6 vaccines-11-00110-t006:** Growth prospects of Indian biosimilars and vaccines industry: factors either contributing positively or affecting negatively.

	Positively Contributing Factors	Negatively Affecting Factors
	Strengths (S)	Weaknesses (W)
**Internal factors**	1. Young and aspiring workforce 2. Cost competitiveness 3. High efficacy, low cost and akin safety level; growing demand in healthcare 4. Affordable, low-cost biosimilars make medication cost-effective in a price-sensitive Indian market 5. Reduced cycle in synthesis and regulatory compliance compared to innovator molecule 6. Innovation, R&D focus in innovative therapeutics as key player at global scale 7. Government regulatory assistance to produce biosimilars 8. Government initiatives to foster confidence and encourage investment	1. Poor Industry-Academia alliance 2. Low government funding to industry 3. Complex regulatory compliance process; lack of confidence in regulatory bodies and policy makers leading to high corporate cost in approval 4. Physicians not prescribing biosimilars; low awareness among the doctor and patient 5. Higher price compared to conventional generic drugs 6. Pharmacovigilance to monitor efficacy and safety needed for possible immunogenicity 7. Altered production process may alter biosimilars’ property 8. Batch-wise uniform production is a challenge; needs skilled manpower, and validated and verifiable SOP
	Opportunities (O)	Threats (T)
**External factors**	1. Green-field, favourable emerging global biosimilars market 2. Fast-growing biopharma trade; US $300 million Indian biosimilar market anticipated to be worth US $40 billion by 2030 3. Over next few years, patent protection of many biologics expire 4. Vast prospect for cost-effective Indian biosimilars; biologics company start-ups booming 5. Biosimilar to cost 20–30% less than biologicals; low cost makes it affordable and accessible as demand grows 6. Government pledges to fund up to US $1.3 billion on API-based pharma business 7. Making APIs locally appear doable in next couple of years; to drive developing biotech-based medicines in India for the world 8. Efforts of DBT and BIRAC to support Indian biotech industry would benefit biologics industry 9. Government strategies focus on globally-acceptable legislation, entrepreneurship, industry-academia and public-private partnerships, and investment avenues for business house, investor and other agencies	1. Stronger Chinese and Korean influence 2. Bargaining power of Indian companies with international patent litigations; patent litigation stifles smaller company from getting into biosimilars business 3. Lack of a comprehensive regulatory framework for biosimilars development 4. Tough approval process for pharama companies to enter global market 5. Delay in clinical trials approval, new pharma pricing policy, uniform code for sales and marketing practises, compulsory licencing, product quality, regularity uncertainty, reluctant to prescribe, and production complexity 6. Substantial competition from branded biologics than the tough competition as posed by small-molecule generics

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
