# Peer review of "Indian Biosimilars and Vaccines at Crossroads–Replicating the Success of Pharmagenerics"

_vaccines, 2023, doi:10.3390/vaccines11010110_

Round 1
Reviewer 1 Report
I think this is a very interesting article. It needs a little bit of review of english in a couple of places but for the most part is very well written and a timely review.
Author Response
Response: The authors extend their thankfulness for the valuable time given and the encouraging observation shared by the learned reviewer.
Reviewer 2 Report
This paper explores Indian biosimilars and vaccines at crossroads. It is an interesting paper. In my opinion, this paper can be accepted for publication after the following revisions:
1. I noticed the “Abstract” has no real structure. I suggest it be rewritten Background, Purpose of Review, Methodology, Results and Conclusions. Right now it is simply just a rambling without any focus.
2. There are a few places where english expression is poor and this needs to be reviewed throughout.
3. I suggest a SWOT analysis section could help the paper to give it structure Strengths, Weakness, Opportunities and Threats to Indian Biosimilars I can see that this would give future directions to research, business and help explain the findings better.
4. I suggest you emphasize the originality of the paper given that it seems to me one of the only ones on the subject material.
Author Response
This paper explores Indian biosimilars and vaccines at crossroads. It is an interesting paper. In my opinion, this paper can be accepted for publication after the following revisions:
- I noticed the “Abstract” has no real structure. I suggest it be rewritten Background, Purpose of Review, Methodology, Results and Conclusions. Right now it is simply just a rambling without any focus.
Response: The observation is appreciated. The Abstract is structured now in the revised manuscript.
- There are a few places where english expression is poor and this needs to be reviewed throughout.
Response: The language issue has been duly addressed in consultation with a language expert all throughout the manuscript.
- I suggest a SWOT analysis section could help the paper to give it structure Strengths, Weakness, Opportunities and Threats to Indian Biosimilars I can see that this would give future directions to research, business and help explain the findings better.
Response: The authors are grateful for the suggestion. A dedicated section along with tabulated details has been added in the revised manuscript.
- I suggest you emphasize the originality of the paper given that it seems to me one of the only ones on the subject material.
Response: The authors agree that the attempt is one of its kinds on the subject matter. In light of this, as per the suggestion, the originality of the paper has been duly highlighted particularly in the Abstract and Conclusion sections in the modified version of the manuscript.
Reviewer 3 Report
The authors have put a lot of effort into this work. They offer exhaustive information on the subject analyzed. However, the structure of the manuscript is not adequate. Every article should have an Introduction in which the work to be conducted is presented, why it is opportune, the objective pursued is established, and the structure of the rest of the work is described.
Likewise, a separate Methodology section should be included describing the methodology used in conducting the research.
Finally, all scientific work has limitations that should be made clear in the final part of the paper.
Author Response
- The authors have put a lot of effort into this work. They offer exhaustive information on the subject analyzed. However, the structure of the manuscript is not adequate. Every article should have an Introduction in which the work to be conducted is presented, why it is opportune, the objective pursued is established, and the structure of the rest of the work is described.
Response: As per the valuable suggestion, a short and succinct Introduction has been added to provide adequate structure to the manuscript.
- Likewise, a separate Methodology section should be included describing the methodology used in conducting the research.
Response: A separate dedicated short ‘Methodology’ section as within the scope of the review paper has been duly included in the revised manuscript. There were certain meticulous planning and execution of the work during sourcing and compiling the secondary data, and further their analysis and synthesis including presenting them in Tables and Figures.
- Finally, all scientific work has limitations that should be made clear in the final part of the paper.
Response: This aspect has been added to the ‘Conclusion’ section in the revised manuscript.
Round 2
Reviewer 3 Report
The recommendations have been addressed by the authors.